# Multimodal Reinforcement Learning with Dynamic Graph Representations for Autonomous Driving Decision-Making

1st Ting Su
*College of Computer and Data Science*
*Fuzhou University*
Fuzhou, China
sting120327@163.com

2nd Chun-Yang Zhang
*College of Computer and Data Science*
*Fuzhou University*
Fuzhou, China
zhangcy@fzu.edu.cn

*Abstract*—It is crucial for autonomous vehicles to make safe and effective decisions in real-time dynamic road environments through decision-making systems. Traditional rule-based decision-making methods struggle to handle complex and variable traffic conditions, limiting their reliability. Data-driven methods such as imitation learning (IL) and reinforcement learning (RL) offer better generalization and adaptability. However, existing methods often fail to capture comprehensive and accurate representations of traffic scenes, especially in high-traffic areas like roundabouts, where neglecting the impact of other traffic participants poses significant safety risks. To address this issue, we propose a multimodal reinforcement learning framework based on integrated dynamic graph representation learning (DGMRL). Our framework designs a spatial-temporal-coupled dynamic graph neural network, which implicitly models temporal information and integrates interaction information and temporal evolution in a unified manner. Additionally, we employ a cross-attention fusion mechanism to effectively integrate multimodal data, constructing a comprehensive driving environment representation. Extensive experimental validation demonstrates that our method outperforms existing baseline models across diverse driving scenarios, with particularly significant performance improvements in traffic-dense environments.

*Index Terms*—Autonomous driving, dynamic graph, decision-making, scene representation, cross-attention

## I. INTRODUCTION

In the advancement of autonomous driving technology, the decision-making module, as the intelligent brain that takes actions based on vehicle perception data, makes its optimization crucial for system performance. With the increasing complexity and variability of the traffic environment, traditional rule-based decision-making methods [1], although stable in specific scenarios, struggle to handle edge cases and sudden changes in real driving conditions, constraining their reliability in the driving process. Algorithms such as imitation learning (IL) [2] and reinforcement learning (RL) [3] [4] , with their ability to learn from data, continuously refine decision-making strategies and demonstrate strong generalization and adaptability.

As the bridge connecting perception and decision-making, the quality of driving environment representation directly determines whether deep reinforcement learning (DRL) can accurately understand complex traffic scenarios and make efficient and safe decisions accordingly. Previous work primarily relied on directly using the input states from perception modules or employing simple processing techniques, such as convolutional neural networks (CNNs), to extract environmental representations [5] [6]. However, the representation of the driving environment not only needs to capture static information such as roads, vehicles, and pedestrians, but also needs to dynamically tracking the changes and interactions among these agents.

In recent years, graphs have been widely applied in driving decision-making due to their efficiency in representing spatial structures within a scene. Some research utilizes graph neural networks (GNNs) to model interactions during driving [7] [8]. However, these approaches are often restricted to static interaction extraction, with some attempts combining recurrent neural networks (RNNs) to capture spatio-temporal features [9]. These methods generally struggle to generalize to unseen graph structures and fail to fully capture the complexities of spatiotemporal co-evolution, leading to limitations in dynamic scene modeling [10].

To address these issues, we design a multimodal reinforcement learning framework based on integrated dynamic graph representation learning(DGMRL) . This framework comprehensively captures spatial and temporal interaction information, ensuring that the decision-making system can understand complex traffic environments in real-time and with high accuracy.Firstly, we developed a spatia-temporal coupled dynamic graph neural network (STC-DGNN), which implicitly considers temporal dependencies and integrates both temporal and topological relationships. Secondly, we deeply mine and effectively integrate complementary information from different modalities to construct a more comprehensive representation of driving environment. Finally, an SAC-based decision-making module with expert strategies is used to generate optimal behavioral decisions. The main contributions are as follows:

- The spatial-temporal-coupled dynamic graph neural network is designed to dynamically adjust GCN parameters. Such adjustment allows model to accurately capture dy-

namic changes in gragh structure.

- A cross-attention fusion mechanism is employed to effectively integrate multi-modal data, and build the DGMRL framework for generating final decisions.
- Through comparisons with existing baseline models, the proposed DGMRL model demonstrate superior performance in various complex driving scenarios.

## II. RELATED WORK

### A. Decision-making Methods for Autonomous Driving

The development of autonomous driving technology relies heavily on efficient decision-making methods. Traditional rule-based decision-making approaches, such as finite state machines (FSM) [1] and decision trees [11], are known for their clear decision logic and strong interpretability. However, these methods depend on predefined rules and logic to guide the decisions of autonomous driving systems, making them inadequate when faced with complex and unpredictable driving scenarios.

Learning-based approaches are able to handle more complex and variable driving environments through data-driven and autonomous exploration. IL improves the training efficiency of complex driving tasks by analyzing extensive human driving data and training networks to simulate expert decision-making behavior [2] [12]. However, the performance of IL models heavily depends on the quantity and quality of expert data, resulting in limited generalization capabilities. RL optimizes strategies by continuous interaction with the environment and using the reward mechanism to optimize the strategy, which makes the autonomous driving system have the ability to make adaptive decisions in uncertain and dynamic environments [3] [4] [13]. Nonetheless, as the complexity of the environment increases, RL requires a large amount of interaction data for experimentation and exploration, which can lead to significant computational resource consumption.

To overcome the challenges, recent research has focused on utilizing a small amount of expert data to guide decisions [6] [14] [15]. This approach cleverly integrates the rapid learning capability of IL with the autonomous exploration ability of RL. Huang et al. [16] guided the policy learning process of DRL by constraining the Kullback-Leibler (KL) divergence between the reinforcement learning policy and the imitated expert policy. However, these methods often overlook the accurate representation of environmental states. In reality, precisely representing the environmental state is crucial for decision models, as it enhances the model's understanding and analysis of driving situations, leading to decisions that better align with real-world scenario demands.

### B. Learning Environmental Representation for Driving

Initially, vehicle driving environment feature representation relied on direct perception module inputs or simple operations like max pooling, pooling, and concatenation to extract environmental information [17]. However, these methods struggle to provide a comprehensive and accurate description of the environment in complex scenarios. Recently, deep learning techniques for extracting environmental features have become mainstream. CNNs are highly effective in image processing and RNNs are advantageous for learning information from historical data [18] [19]. Nevertheless, these traditional approaches still fall short in capturing the interactions among traffic participants, failing to grasp geometric information and semantic connections in the environment. Especially in congested areas such as roundabouts and toll booths, ignoring the impact of other road users can be very dangerous.

Due to its unique advantages in handling complex network structured data, GNNs have been widely applied in the field of autonomous driving [8] [20] [21]. By abstracting various elements in traffic scenarios, such as vehicles, pedestrians, and road nodes, into graph nodes and representing their interactions as edges [7], GNNs leverage their unique information propagation and aggregation capabilities to effectively capture the relationships between traffic participants and more accurately understand and predict their behavior [22]. Qi Liu et al. [23] used GCNs for interaction feature extraction and combined it with four different DRL methods to generate collaborative decisions. Peide Cai et al. [24] employed GATs to encode heterogeneous traffic information and performed end-to-end training of the decision model using Q-learning.

However, as vehicles move on the road, the interactions between traffic participants continually evolve over time. Static graphs alone cannot capture these dynamic interactions and overlook temporal dependencies in the driving process [25] [26]. Xuemin Hu et al [9] used RNNs to temporally encode graph embedding in order to achieve learning the dynamic relationships between nodes in a changing driving scenario. However, these networks that only encode temporal information for interactions are unable to effectively extract features for unseen nodes [27] and treat temporal and topological information separately, ignoring the synchronized evolution of vehicle interactions over time. To address this limitation, we designed STC-DGNN to integrate temporal information into the model parameters, significantly enhancing the ability to extract features for newly appearing vehicle. Additionally, we used a cross-attention fusion mechanism to merge data from different modalities, and developed a multi-modal reinforcement learning framework based on integrated dynamic graph.

## III. METHODOLOGY

The proposed DGMRL framework, as illustrated in Fig. 1, aims to derive a comprehensive and precise state representation that captures the current vehicle interactions, motion trends, and road conditions. This state representation will serve as the core basis for driving decisions, enabling the generation of more accurate driving actions. The following sections will elaborate on the key modules of this framework and their respective functions.

### A. Integrated Dynamic Graph Neural Network

We design the STC-DGNN module, which uses a recurrent model to evolve the parameters during the graph convolution

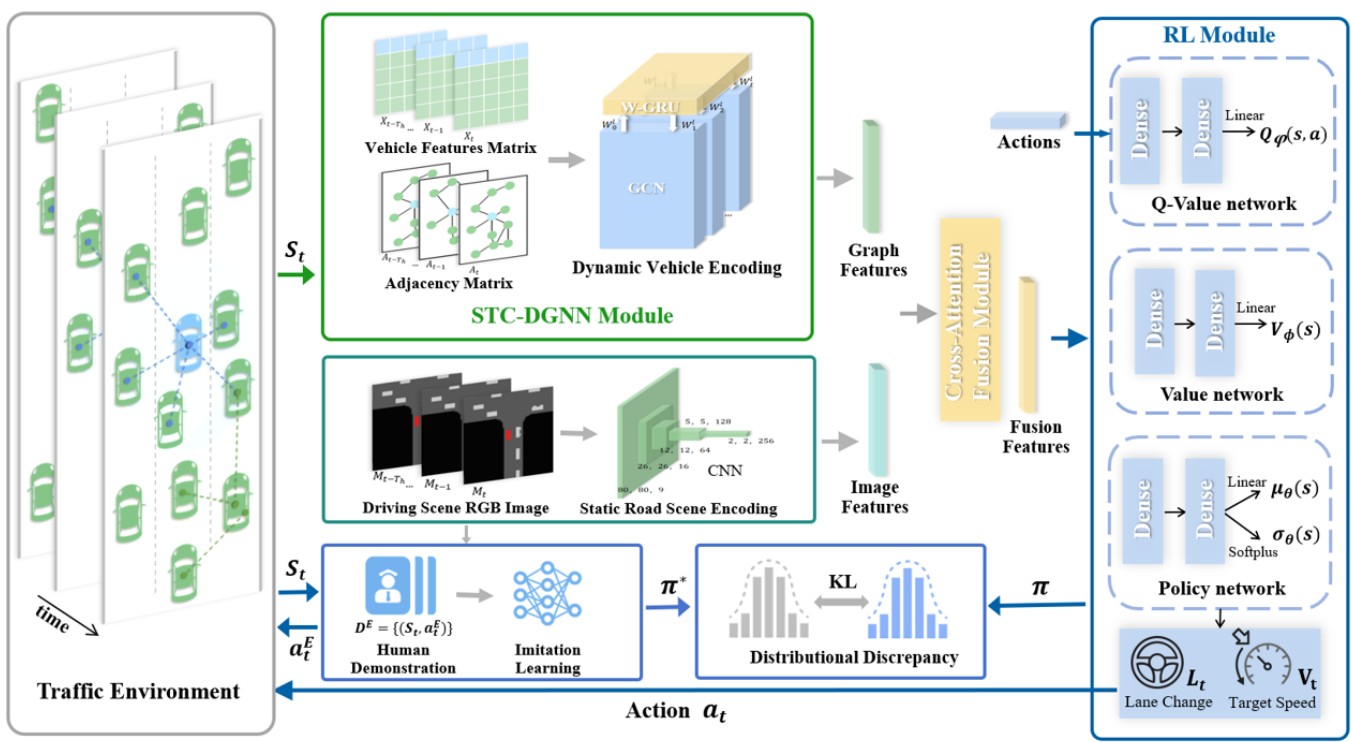

Fig. 1: Multimodal Reinforcement Learning with Integrated Dynamic Graph Representations Learning (DGMRL) framework for Decision-making of Autonomous Driving.

process, thereby capturing the dynamic nature of the graph structure underlying vehicle interactions.

At time $t$, a set of ordered historical state graphs $\mathcal{G} = \{G_{t-T_h+1}, G_{t-T_h+2}, \ldots, G_t\}$ is input, with $T_h$ as the trace-back horizon. At each time step $t$, $G_t = (V_t, E_t)$ includes the set of vehicle nodes and the set of interaction edges. The vehicle feature matrix $X_t \in \mathbb{R}^{N \times f}$ represents the features of the $N$ vehicles, where $f$ is the number of features for each node. The adjacency matrix $A_t \in \mathbb{R}^{N \times N}$ represents the interactions between vehicles. Most existing works directly consider all vehicles within a distance threshold as neighbors, without distinguishing the importance of their influence, which does not align with real-world rules. To better differentiate the influence of different neighboring vehicles on the target vehicle, we assign weights to the edges in $E_t$, with the weighted adjacency matrix formulated as

$$A_t = \left\{ a_t^{ij} \right\}_{i,j=1}^{N}, \tag{1}$$

$$a_t^{ij} = \exp\left(-\frac{d_t^{ij}}{\tau}\right), \tag{2}$$

where $d_t^{ij}$ represents the distance between two nodes and $\tau$ is the temperature coefficient.

The spatial-temporal graph contains raw information about the dependencies between vehicles. We need an effective strategy to obtain high-quality representations of vehicle information. Most current methods use RNNs to sequentially connect node embeddings generated by GCN, considering spatial and temporal dimensions separately, which is not well-suited for the frequent appearance and disappearance of vehicles. The core of STC-DGNN is to continuously update the GCN weight parameters based on current and historical information, utilizing weights to store historical time information [28]. This better simulates the changes of graph nodes and edges over time, considering spatiotemporal information in a unified manner. We use a GRU to update the weights at each time step. The GCN weights $w_{t-1}^{(l)}$ generated at the previous time step serve as the input to the GRU, generating the weights for the next time step $w_t^{(l)}$, which can be defined as

$$w_t^{(l)} = \text{GRU}(w_{t-1}^{(l)}, H_t^{(l)}), \tag{3}$$

where $H_t^{(l)}$ is the vehicle node embedding of the $l$-th layer network at time $t$. The generated $w_t^{(L)}$ serves as the new weights for the GCN. The node embedding matrix $H_t^{(l)}$ and the adjacency matrix $A_t$ are used as inputs to update the node embeddings at the current time step, integrating more information from nearby vehicles:

$$\begin{aligned} H_t^{(l+1)} &= \text{GCN}(H_t^{(l)}, A_t) \\ &= \sigma\left(\hat{D}_t^{-1/2} \hat{A}_t \hat{D}_t^{-1/2} H_t^{(l)} W_t^{(l)}\right), \end{aligned} \tag{4}$$

where $\sigma$ is the activation function (usually ReLU), $\hat{A}_t = A_t + I$ is the adjacency matrix with added self-loops, and $\hat{D}_t$ is the diagonal degree matrix of $\hat{A}_t$. The initial embedding matrix comes from the node features, i.e., $H_t^{(0)} = X_t$. It is worth noting that in our model, the GCN parameters are updated only by the GRU and are no longer trained. This parameter update mechanism allows the model to more effectively utilize historical information, reduce computational complexity, and improve training efficiency. Furthermore, as our method captures dynamics by adjusting GCN weights at each time step, we do not require graph structure data for the entire time span, making the model more suitable for real-world scenarios.

### B. Cross-Attention for Multi-Modal Fusion

We not only focus on the topological relationships and temporal dynamics between traffic participants but also on the efficient encoding and understanding of the overall road structure. Using upstream perception modules, we sample bird's-eye view images from historical moments to simulate real-world highway environments as closely as possible. The images are abstracted into RGB format, where roads are marked in gray, the ego vehicle is highlighted in red, and other traffic participants are shown in white. This provides an intuitive way to quickly identify and distinguish different road elements, forming an information-rich scene representation. We use four convolutional layers and a global average pooling layer to effectively capture details of the road layout and the relative positions of vehicles in the traffic flow, generating features that reflect traffic conditions and the ego vehicle's position.

Compared to traditional concatenation methods, as shown in Fig. 2 , we employ cross-attention [29] to fuse embeddings from different modalities by computing the correlation between image and dynamic graph features. This approach intelligently selects the most critical information from each modality for the current decision.

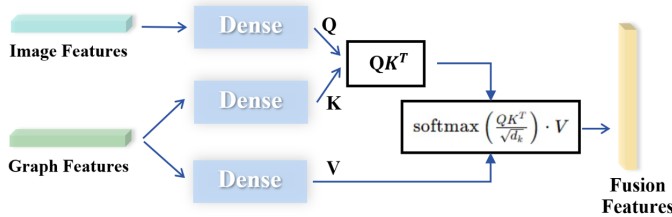

Fig. 2: Using cross-attention to fuse information from two modalities.

Specifically, the image features encoded by the CNN serve as the Query, while the dynamic graph features encoded by the STC-DGNN act as the Key and Value. This mechanism can be expressed as

$$\text{Attention}(Q, K, V) = \text{softmax}\left(\frac{QK^T}{\sqrt{d_k}}\right) V, \quad (5)$$

where $d_k$ is the square root of the feature dimension used as a scaling factor.

The multi-head Attention mechanism captures different aspects of the information by using multiple attention heads. Each head computes the attention-weighted values given the Query matrix $Q$, Key matrix $K$, and Value matrix $V$, which are transformed by the corresponding weight matrices $W_i^Q$, $W_i^K$, and $W_i^V$:

$$\text{MultiHead}(Q, K, V) = \text{Concat}(\text{head}_1, \ldots, \text{head}_H) W^R, \quad (6)$$

$$\text{head}_i = \text{Attention}(QW_i^Q, KW_i^K, VW_i^V), \quad (7)$$

where $\text{head}_i$ represents the output of the $i$-th attention head, and $W^R$ is the final output weight matrix used to map the concatenated multi-head output back to the original feature dimension.

Through this approach, the model can more precisely process and integrate information from different modalities, reducing redundancy and enhancing the relevance of feature representations. This ultimately provides a richer and more accurate environmental feature representation for the autonomous driving decision-making system.

### C. Reinforcement Learning with Expert Priors

We consider the vehicle decision-making problem as a Markov Decision Process (MDP), defined as $M = (S, A, T, R, \gamma)$. The objective is to derive an effective policy $\pi(a_t \mid s_t)$ that selects suitable actions based on real-time observations at each step, with the goal of maximizing the long-term discounted cumulative reward to achieve optimal decision-making. Here, $S$ represents the state space, $A$ is the action space, $T : S \times A \rightarrow S$ is the transition probability distribution, $R$ rewards efficient paths and safe driving, and $\gamma$ adjusts the significance of future rewards.

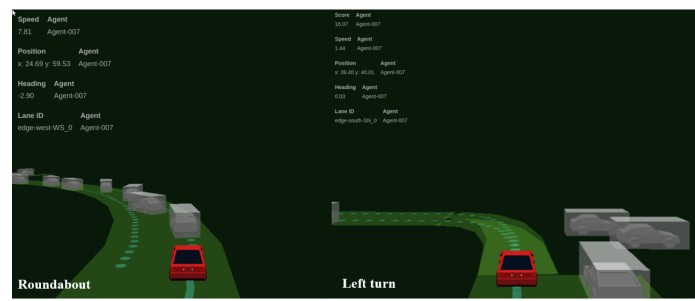

Fig. 3: Experts drive the vehicle from a first-person perspective and record state-action pairs.

**1) Imitative Expert Policy:** We create a simulated expert policy using behavioral cloning by collecting human driving data. Human experts, in identical driving scenarios (e.g., as illustrated in Fig. 3 , adjust vehicle behavior based on real-time states using their practical experience. The actions include four distinct maneuvers: acceleration, deceleration, lane change to the left, and lane change to the right. All state values are normalized to the range [0, 1], while action values are normalized to the range [-1, 1].

The successful task completion trajectories are recorded as $P = \{\rho_i\}_N$. Expert data is documented as state-action pairs

$\rho = \{(s_1, a_1), \ldots, (s_T, a_T)\}$. The expert policy $\pi^* : S \to A$ is obtained by maximizing the log-likelihood over $P$ as

$$\pi^* = \underset{\pi}{\arg\max}\ \mathbb{E}_{(s,a) \sim P}\left[\log \pi(a \mid s)\right]. \tag{8}$$

We train the expert policy using a neural network by minimizing the mean squared error between the neural network's action output and the expert actions, optimizing the expert policy network, shown as

$$L(\theta) = \mathbb{E}_{(s,a) \sim P}\|\pi_\theta(s) - a\|^2. \tag{9}$$

Since human driving behavior has strategic uncertainty, different driving actions may be taken in the same scenario. Therefore, we assume that actions follow a Gaussian distribution $a_t \sim \mathcal{N}(\mu_\theta(s_t), \sigma_\theta^2(s_t))$, rather than being deterministic. The imitation policy is trained using maximum likelihood estimation [30]

$$L(\theta) = \mathbb{E}_{(s,a) \sim P}\left[\frac{\log \hat{\sigma}_\theta^2(s)}{2} + \frac{(a - \hat{\mu}_\theta(s))^2}{2\hat{\sigma}_\theta^2(s)} + d\right], \tag{10}$$

where $\theta$ denotes the parameters of the policy network, $\hat{\mu}_\theta$ and $\hat{\sigma}_\theta^2$ are the predicted mean and variance, and $d$ is a constant.

However, in practical driving scenarios, there are many situations not covered by the expert data, leading to uncertainties and unreliability in the predicted mean and variance. This model uncertainty arises from insufficient training data in certain regions of the state space. We estimate this uncertainty by training $K$ networks with different random initializations and data orders, combining the results into a Gaussian mixture distribution [31]. The mixture mean and variance are calculated as

$$\mu_{\pi^*}(s) = \frac{1}{K}\sum_{i=1}^{K} \hat{\mu}_{\theta_i}(s), \tag{11}$$

$$\sigma_{\pi^*}^2(s) = \frac{1}{K}\sum_{i=1}^{K} \hat{\sigma}_{\theta_i}^2(s) + \left[\frac{1}{K}\sum_{i=1}^{K} \hat{\mu}_{\theta_i}^2(s) - \mu_{\pi^*}^2(s)\right], \tag{12}$$

where $\theta_i$ represents the parameters of the $i$-th network.

**2) Soft Actor-Critic (SAC) [32]:** We utilize cross-attention fused dynamic graph and image features as inputs for the state features in reinforcement learning. The SAC algorithm is used to output autonomous driving decision behaviors. SAC introduces maximum entropy on top of maximizing future cumulative rewards, thereby maximizing the cumulative reward while ensuring the policy is as random as possible, allowing each action to have a certain output probability, which improves the agent's exploration ability. This algorithm learns four networks simultaneously: two Q-function networks $(Q_{\varphi_1}, Q_{\varphi_2})$, one value function network $(V_\phi)$, and one stochastic policy network $(\pi_\theta)$. It also retains a target value network to stabilize the training of the Q-function networks. The loss functions for the Q-function networks are identical, both minimizing the Mean Squared Bellman Error (MSBE) function, given by

$$\mathcal{L}(tashuo\varphi_i) = \mathbb{E}_{(s_t,a_t,r_t,s_{t+1}) \sim D}\big[(Q_{\varphi_i}(s_t, a_t) \\ - (r_t + \gamma V_{\bar{\phi}}(s_{t+1})))^2\big], \tag{13}$$

where $D$ is the experience replay buffer. The value function is updated via the following loss function:

$$\mathcal{L}(\phi) = \mathbb{E}_{\substack{s_t \sim D \\ \hat{a}_t \sim \pi_\theta(\cdot|s_t)}}\left[\frac{1}{2}\left(V_\phi(s_t) - \min_{i=1,2} Q_{\varphi_i}(s_t, \hat{a}_t)\right)^2\right], \tag{14}$$

where the action $\hat{a}_t$ is newly sampled from the policy network, and the state $s_t$ comes from the experience replay buffer. When updating the value function, only the smaller Q-value is used to mitigate overestimation of the value function. The policy network $\pi_\theta$ outputs the mean $\mu$ and standard deviation $\sigma$, with the final action sampled from a Gaussian distribution $a_t \sim \mathcal{N}(\mu_\theta(s_t), \sigma_\theta^2(s_t))$. The loss function for the policy network is

$$L(\theta) = \mathbb{E}_{\substack{s_t \sim D \\ \hat{a}_t \sim \pi_\theta(\cdot|s_t)}}\left[-\min_{i=1,2} Q_{\varphi_i}(s_t, \hat{a}_t)\right]. \tag{15}$$

**3) Learning With Expert Policy:** We introduce the KL divergence to describe the discrepancy between the policy network $\pi_\theta$ and the imitation expert policy $\pi^E$, which is formulated as

$$D_{\mathrm{KL}} = \mathrm{KL}\left[\pi_\theta(\cdot \mid s_t) \,\|\, \pi^*(\cdot \mid s_t)\right], \tag{16}$$

this allows the agent's policy to be regularized towards the expert policy. Thus, the reconstructed reward function is given by

$$r'(s_t, a_t) = r(s_t, a_t) - \alpha D_{\mathrm{KL}}, \tag{17}$$

where $\alpha$ is a temperature parameter that measures the importance of the KL divergence. By incorporating $D_{\mathrm{KL}}$ as a penalty term into the value function, the loss function is modified as

$$\mathcal{L}(\phi) = \mathbb{E}_{\substack{s_t \sim D \\ \hat{a}_t \sim \pi_\theta(\cdot|s_t)}}\left[\left(V_\phi(s_t) - \left(\min_{i=1,2} Q_{\varphi_i}(s_t, \hat{a}_t) - \alpha \hat{D}_{\mathrm{KL}}\right)\right)^2\right], \tag{18}$$

where $\hat{D}_{\mathrm{KL}}$ is the estimated KL divergence, The loss function for the policy network is updated to

$$\mathcal{L}(\theta) = \mathbb{E}_{\substack{s_t \sim D \\ \hat{a}_t \sim \pi_\theta(\cdot|s_t)}}\left[-\min_{i=1,2} Q_{\varphi_i}(s_t, \hat{a}_t) + \alpha \hat{D}_{\mathrm{KL}}\right]. \tag{19}$$

By regularizing the learned policy $\pi_\theta$ to the expert policy $\pi^E$, we use expert priors a combined with RL methods to help the decision system solve autonomous driving tasks safely and efficiently.

## IV. EXPERIMENT

### A. Implementation details

During training, our method and other baseline methods are each trained 5 times using different random seeds, with 200,000 steps per training session. The simulation interval is set to 0.1 seconds, and the learning rate is 5e-5. Model checkpoints are saved based on the average return. All networks are trained using TensorFlow and the Adam optimizer on an NVIDIA GeForce RTX 3060 GPU.

For testing, the saved policy networks are evaluated over 200 episodes, with success rate serving as the evaluation metric. Success rate is defined as the ratio of episodes that successfully reach the goal without collisions within the given step limit to the total number of episodes.

## B. Driving Scenarios Settings

Our method is trained and tested on the SMARTS simulation platform [33]. SMARTS provides highly customizable traffic environments and a range of simulation tools to model real-world driving conditions and vehicle interactions. We designed two different scenarios: a left turn and a roundabout, as shown in Fig. 4 . The autonomous vehicle must navigate these scenarios safely and efficiently based on the intentions of other vehicles and road conditions.

1) In the left-turn scenario, the self-driving vehicle needs to turn left at a T-shaped intersection without traffic lights in a busy city, and the ego vehicle needs to turn left from the main road into the secondary road.

2) The roundabout scene simulates the behavior of vehicles when entering, driving and leaving the roundabout. By setting different traffic flow types, traffic flow rates, and Agent target tasks, three different difficulty traffic scenes are designed, divided into easy, medium and hard.

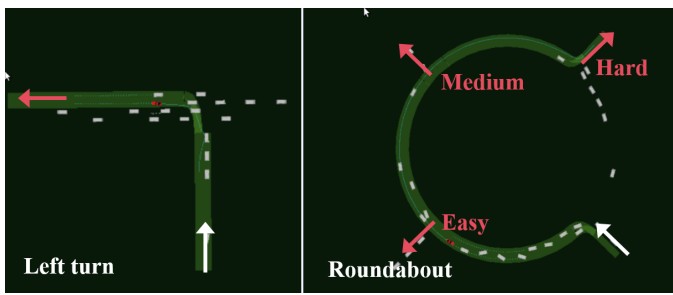

Fig. 4: The white arrow shows the starting point, and the red arrow shows the destination. The images show left-turn and roundabout scenarios. The roundabout are classified into three difficulty levels based on different exits and traffic density.

## C. Decision-making Process Settings

**1) State Space:** The state space is divided into two complementary dimensions: Dynamic Graph and RGB Image. This comprehensive approach captures the rapidly changing driving environment, ensuring that the agent can make precise decisions based on multimodal information. The Dynamic Graph uses snapshot data from the past 7 time steps, where each snapshot includes a feature matrix detailing each vehicle's state attributes at that time, as well as an adjacency matrix describing the relative positions and interactions between vehicles. The RGB Image module focuses on a 32×32 meter field of view centered on the ego vehicle, with a resolution of 0.4 meters per pixel, ensuring accurate detail capture. RGB images from the past 3 time steps are used to form an 80×80×9 input matrix. By incorporating historical time steps, the State Space design records not only static information but also enables the agent to identify environmental change trends through temporal comparison, aiding reinforcement learning in making predictive decisions.

**2) Action Space:** In this work, the agent performs speed and lane changes in the action space as $a_t = [V_t, L_t]$. The agent controls the vehicle based on road congestion and driving rules. Longitudinal movement is controlled by speed $V_t$, which ranges from 0 to 10 m/s. Lateral control consists of discrete lane change actions $L_t$, where $L_t = \pm 1$ represents left/right lane change and $L_t = 0$ indicates maintaining the current lane. To effectively handle the composite action space containing both continuous and discrete actions, the action values are normalized to the range of $[-1, 1]$ to avoid potential biases or imbalances arising from the differences in action space characteristics.

**3) Reward Function:** We use a sparse reward function consisting of a collision penalty and a goal achievement reward, mathematically expressed as $R(s, a) = r_{\text{collision}} + r_{\text{achievement}}$, where $r_{\text{collision}} = -1$ represents the negative reward received when the agent collides with other traffic participants in the environment, and $r_{\text{achievement}} = 1$ represents the positive reward received when the agent successfully reaches the designated goal position. If the goal is not reached within the specified number of steps, the reward function returns 0. This design guides the agent's learning process by clearly distinguishing between success and failure states, thus optimizing its decision-making strategy.

## D. Baselines

To compare the feasibility of the proposed method, we evaluate it against both classical decision-making methods and state-of-the-art approaches. The baseline methods are as follows:

1) **Generative Adversarial IL (GAIL)** [34]: GAIL is an algorithm that combines Generative Adversarial Networks (GANs) with Reinforcement Learning (RL). It optimizes its policy by maximizing a reward signal that reflects similarity to expert data, enabling efficient policy learning and behavior imitation.

2) **Proximal Policy Optimization (PPO)** [35]: PPO is an efficient policy optimization algorithm that employs Convolutional Neural Networks (CNNs) to encode image inputs for decision-making. By constraining the magnitude of policy updates, it balances exploration and exploitation, particularly excelling in complex decision problems requiring fine-grained long-term planning.

3) **Soft Actor-Critic (SAC)** [32]: SAC is a baseline RL method that operates without scene understanding, which combines the principles of maximum entropy reinforcement learning and the actor-critic framework. The method directly receives state input as representation but uses LSTM for encoding to gather information at each step.

4) **Expert-Prior-RL (EPRL)** [16]: This recent innovative RL framework combines expert demonstrations and policy cloning, utilizing KL divergence regularization to promote DRL learning. It exhibits highly successful, human-like driving behavior and similarly uses image-based environmental representations.

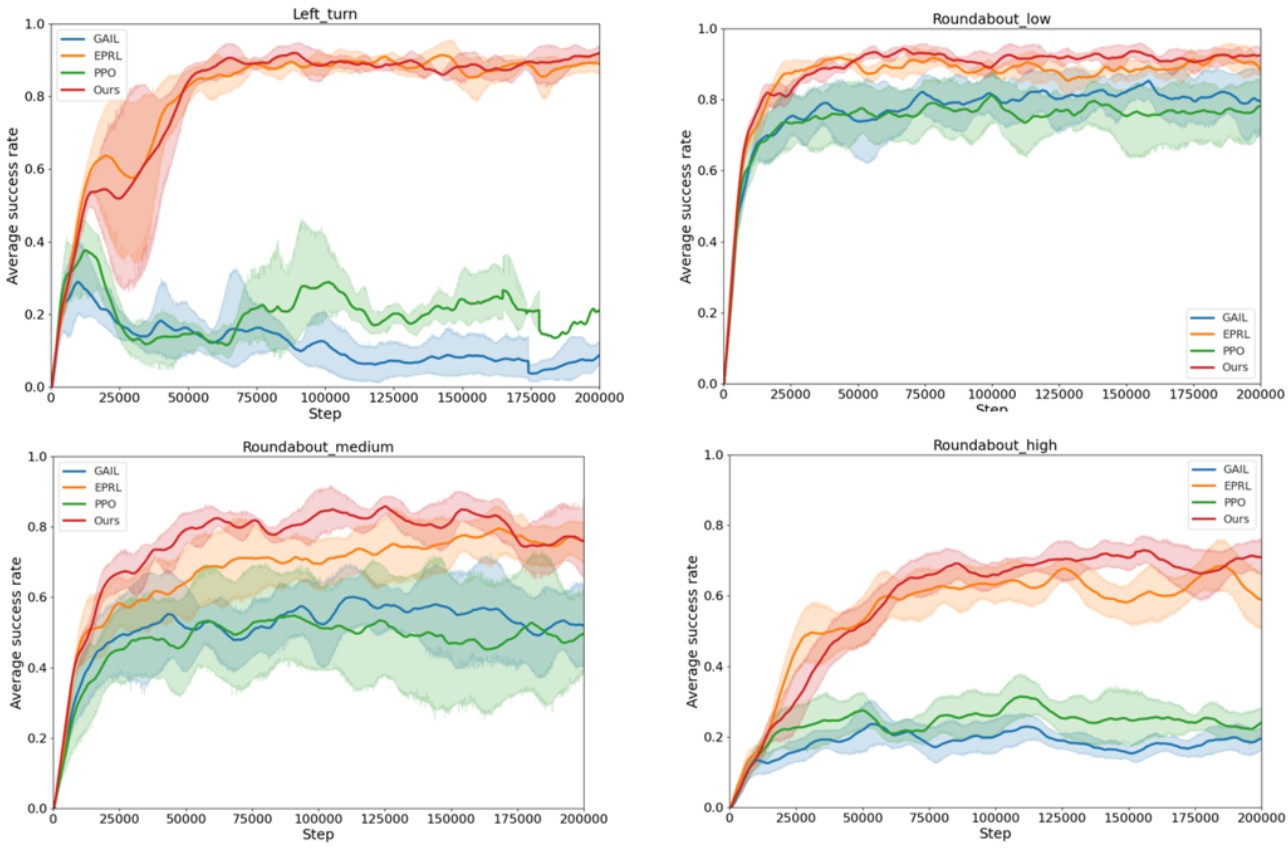

Fig. 5: The average success rate during training in four different driving scenarios.

### E. Training Results

We traine the DGMRL method and the baseline method in four different driving scenarios designed by us and the training curves are shown in Fig. 5 . In order to ensure fair comparison, we use the average success rate as the training metric. The average success rate of each step is the mean success rates of the historical 20 episodes. Meanwhile, the solid line in the figure is the average of multiple training curves, and the color band is the standard deviation error of multiple trainings. Moreover, We use each training curve to represent the average (solid line) and standard deviation error band of the method to obtain the average training success rate. The training curves show that our method has the best training performance in four different driving scenarios. In the left-turn scenario and the simple roundabout scenario, our method has an average success rate of more than 90% after a certain number of steps and remains stable throughout the training process, which is much higher than the two classic reinforcement learning algorithms and slightly higher than EPRL. When the autonomous driving vehicle appears in the two more complex switching scenarios, our advantage becomes more prominent, and the training curve after convergence is significantly higher than Baselin. The proposed method achieves efficient and accurate representation of complex traffic scenes by fusing two modal data, dynamic

graph and image, in a cross-attention mechanism. Therefore, in more dynamic and complex scenarios, more accurate decisions can be made and the occurrence of collisions caused by wrong decisions can be reduced. Such experimental results show that our method has strong adaptability and stability in driving scenarios of different complexities and is the most effective strategy.

### F. Testing Results

To better evaluate DGMRL 's robustness, we empirically test in four scenarios that are not accessible during training. Specifically, we report the success rate as experimental results. Table I illustrates that our proposed DGMRL framework outperforms existing reinforcement learning methods. While the conventional GAIL and PPO approaches demonstrate poor performance in all scenarios we proposed, employing SAC to autonomous driving has significantly improved decision-making capabilities. Futhermore, SRT has improved the success rate by effectively encoding driving scenarios based on the SAC algorith. Additionally, EPRL has a better understanding of the driving environment through CNN and introduces the imitation expert strategy to promote the improvement of RL. Our method exhibits performance comparable to EPRL in left-turning and simple lane-changing tasks, maintaining a

TABLE I: Testing Results in Four Different Scenarios

| Method | Left_turn | Roundabout_easy | Roundabout_medium | Roundabout_hard |
|--------|-----------|-----------------|-------------------|-----------------|
| GAIL [34] | 28.5% | 85.5% | 70.0% | 33.0% |
| PPO [35] | 48.0% | 81.5% | 68.0% | 39.0% |
| SAC [32] | 86.5% | 87.0% | 81.5% | 73.5% |
| EPRL [16] | 95.5% | 93.0% | 89.0% | 87.0% |
| **OURS** | **96.5%** | **98.0%** | **93.5%** | **93.5%** |

high level of decision-making capability with some minor improvements. An even more exciting outcome of our study is that CFL show superior improvement in the busier roundabout scenarios at medium and hard levels, highlighting that our method effectively solves the low success rate learning models in busy and crowded traffic conditions. This is mainly due to our dynamic graph representation learning, which can cleverly capture the motion trends and interactions of surrounding vehicles in traffic congestion. Such analysis shows that our algorithm is not only capable of handling simple and medium-complexity tasks, but also maintains excellent performance when dealing with highly complex and difficult environments.

### G. Ablation Studies

Our ablation experiment includes the following two parts: the first ablation study compares the performance of different graph encoders on the tasks of left-turning and navigating a complex roundabout scenario. As shown in Table II, the results clearly indicate that the STC-DGNN significantly outperforms both the standard GCN and the GCN combined with GRU in the dense roundabout scenario.

TABLE II: Comparison of Testing Performance for Different Graph Representation Learning Methods

| Ablation | Left_turn | Roundabout_hard |
|----------|-----------|-----------------|
| GCN | 94.5% | 88.0% |
| GCN+GRU | 93.0% | 91.0% |
| STC-DGNN | 96.5% | 93.5% |

The other ablation study focuses on different methods of fusing multimodal data, as illustrated in Table III. The results demonstrate that the attention mechanism significantly outperforms a simple concatenation approach in both scenarios, particularly in the more complex roundabout task, highlighting the attention mechanism's ability to effectively integrate and weigh different data sources.

TABLE III: Comparison of Testing Performance for Different Multimodal Data Fusion Methods

| Ablation | Left_turn | Roundabout_hard |
|----------|-----------|-----------------|
| Concat | 96.00% | 88.00% |
| Attention | 96.50% | 93.50% |

## V. CONCLUSIONS AND FUTURE WORK

This paper proposes DGMRL to address the challenges faced by autonomous driving decision systems in complex and dynamic traffic environments. We have designed a spatial-temporal coupled dynamic graph neural network that can implicitly model temporal dependencies and dynamically adjust GCN parameters, excelling in capturing dynamic changes in graph structures. Additionally, we introduce a cross-attention fusion mechanism to further enhance the integration and balancing of multimodal data, significantly improving the performance of the decision-making system in complex driving scenarios. Comparative experiments with existing baseline models demonstrate that our method achieves superior performance across various challenging driving scenarios, with particularly notable improvements in traffic-dense environments. Future research could explore more diverse multimodal data fusion strategies to enhance the model's robustness under extreme conditions. Furthermore, we plan to expand the scale and diversity of driving scenarios to validate the model's generalization and adaptability in broader applications.

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
