# OpenReview forum: "Multimodal Reinforcement Learning with Dynamic Graph Representations for Autonomous Driving Decision-Making"
_IEEE.org/ICIST/2024/Conference — IEEE ICIST 2024 Conference Submission_

### Official Review · Reviewer_6T3T · 2024-08-21
**This manuscript has a certain degree of innovation and clear simulation figures. It is recommended to accept this paper for publication in IEEE ICIST 2024.**

**Rating:** 7
**Confidence:** 4

**Review:**

This manuscript has a certain degree of innovation and clear simulation figures. Please reply to the following questions one by one.

The text mentions that the spatial-temporal-coupled dynamic graph neural network is designed to dynamically adjust GCN parameters. Can you elaborate on the specific mechanisms employed to achieve this dynamic adjustment? How does the model detect changes in the graph structure, and how does it adapt its parameters accordingly?

The paper claims that the proposed DGMRL model demonstrates superior performance compared to existing baseline models in various complex driving scenarios. Could you expand on the experimental setup and results? Specifically, which baseline models were used for comparison? What metrics were employed to evaluate performance?

---

### Official Review · Reviewer_TjKG · 2024-08-21
**Good paper**

**Rating:** 7
**Confidence:** 3

**Review:**

This article proposes a multimodal reinforcement learning framework with dynamic graph representations to enhance decision accuracy and adaptability for autonomous driving in complex traffic environments. This paper is well organized and contains meaningful results. The following comments should be considered in the revision.
1.The proposed multimodal reinforcement learning framework utilizing dynamic graph representations has demonstrated remarkable performance improvements in complex traffic environments. To further strengthen the model's generalization capabilities and prepare it for real-world deployment, it would be beneficial to expand the diversity of testing scenarios. This could include highway driving, urban street navigation, mountainous terrain traversal, and driving in adverse weather conditions. Testing across these various scenarios would validate the model's robustness and adaptability, fostering its potential for broader commercial applications.
2.The cross-attention mechanism employed in this work effectively fuses multimodal data, showcasing promising results. To elevate data integration efficiency and accuracy, future research could delve into more advanced fusion techniques. For instance, integrating adaptive attention weighting mechanisms or leveraging graph neural networks and graph attention networks could offer finer-grained feature extraction and data integration. These enhancements would empower the model to make precise decisions in even more complex and high-dimensional driving environments.
3.The safety of autonomous driving systems is paramount. While the experimental results highlight the model's high success rate across various scenarios, further intensifying safety assessments and validations is crucial. Future endeavors should incorporate evaluations of the model's performance under extreme conditions, such as emergency avoidance maneuvers and vehicle control failures. Combining hardware-in-the-loop simulations with real-world road tests would comprehensively verify the model's safety, ensuring reliable operation in real-life applications.

---

### Official Review · Reviewer_8eeF · 2024-08-22
**Manuscript Accept**

**Rating:** 7
**Confidence:** 4

**Review:**

What is the purpose of the imaginary white lines in Fig.4?
How do the authors define the medium and hard difficulty in the roundabout scene?
What is the rationale for dynamically adjusting GCN parameters through the spatial-temporal-coupled dynamic graph neural network?
Scattered throughout this manuscript, some format errors should be corrected.

---

### Decision · Program_Chairs · 2024-09-08

Accept (Oral)